**METHOD**

# Bento: a toolkit for subcellular analysis of spatial transcriptomics data

Clarence K. Mah[1,2,3†], Noorsher Ahmed[2,3†], Nicole A. Lopez[2], Dylan C. Lam[2,3,4], Avery Pong[2], Alexander Monell[2,5], Colin Kern[6], Yuanyuan Han[6], Gino Prasad[2,5], Anthony J. Cesnik[7], Emma Lundberg[7,8,9], Quan Zhu[6], Hannah Carter[1] and Gene W. Yeo[2,3,10,11*]

†Clarence K. Mah and Noorsher Ahmed are co-first authors and contributed equally.

*Correspondence:
geneyeo@ucsd.edu

[1] Division of Medical Genetics, Department of Medicine, University of California San Diego, La Jolla, CA, USA
[2] Department of Cellular and Molecular Medicine, University of California San Diego, La Jolla, CA, USA
[3] Sanford Stem Cell Institute Innovation Center, La Jolla, CA, USA
[4] Division of Biological Sciences, University of California San Diego, La Jolla, CA, USA
[5] Department of Bioengineering, University of California San Diego, La Jolla, CA, USA
[6] Center for Epigenomics, University of California San Diego, La Jolla, CA, USA
[7] Department of Bioengineering, Stanford University, Stanford, CA, USA
[8] Department of Pathology, Stanford University, Stanford, CA, USA
[9] Chan-Zuckerberg Biohub, San Francisco, CA, USA
[10] Stem Cell Program, University of California San Diego, La Jolla, CA, USA
[11] Institute for Genomic Medicine, University of California San Diego, La Jolla, CA, USA

## Abstract

The spatial organization of molecules in a cell is essential for their functions. While current methods focus on discerning tissue architecture, cell–cell interactions, and spatial expression patterns, they are limited to the multicellular scale. We present Bento, a Python toolkit that takes advantage of single-molecule information to enable spatial analysis at the subcellular scale. Bento ingests molecular coordinates and segmentation boundaries to perform three analyses: defining subcellular domains, annotating localization patterns, and quantifying gene–gene colocalization. We demonstrate MERFISH, seqFISH+, Molecular Cartography, and Xenium datasets. Bento is part of the open-source Scverse ecosystem, enabling integration with other single-cell analysis tools.

## Introduction

The spatial organization of molecules in a cell is essential for performing their functions. While protein localization [1] and disease-associated mislocalization are well appreciated [2, 3], the same principles for RNA have begun to emerge. For instance, the spatial and temporal regulation of RNA play a crucial role in localized cellular processes such as cell migration and cell division [4, 5], as well as specialized cell functionalities like synaptic plasticity [6–8]. Mislocalization of RNA has been associated with diseases such as Huntington's disease (HD), where defects in axonal mRNA transport and subsequent translation in human spiny neurons lead to cell death and neurodegeneration [9–12].

The study of subcellular RNA localization necessitates single-molecule measurements. Since the development of single-molecule fluorescent in situ hybridization (smFISH), recent advances in multiplexed methods such as MERFISH [13], seqFISH+[14], HybISS [15], and Ex-Seq [16] have enabled RNA localization measurements at near transcriptome scales, while maintaining single-molecule resolution. A number of computational toolkits, such as Squidpy [17], stLearn [18], Giotto [19], Seurat [20], and Scanpy [21]

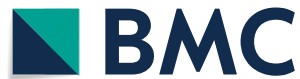

enabled the characterization of tissue architecture, cell–cell interactions, and spatial expression patterns. Despite the single-molecule measurements in spatial transcriptomics, these analytical approaches are limited to investigating spatial variation at the multicellular scale and lack the ability to investigate subcellular organization. To further our understanding of RNA localization and its function in normal and abnormal cell activity, we need to expand our analytical capacity to the subcellular scale.

Recent methods such as FISH-quant-v2 [22] and FISHFactor [23] identify subcellular patterns describing the spatial distribution of RNA species, but are unable to annotate more than a single gene per cell or are limited to analyze at most 20,000 molecules on accessible computing resources. In contrast, a single spatial transcriptomics experiment measures at least hundreds to thousands of genes across hundreds of thousands of cells. Additionally, methods such as ClusterMap [24] and Baysor [25] highlight the potential for transcript locations alone to inform meaningful domains such as cell and nuclear regions. Using spatial proteomics data, CAMPA [26] and Pixie [27] utilize subcellular spatial variation in protein abundance to identify subcellular regions and annotate pixel-level features.

Building on these promising approaches, we present Bento, an open-source Python toolkit for scalable analysis of spatial transcriptomics data at the subcellular resolution. Bento ingests single-molecule resolution data and segmentation masks, utilizing geospatial tools (GeoPandas [28], Rasterio [29]) for spatial analysis of molecular imaging data, and data science tools including SciPy [30] and Tensorly [31] for scalable analysis of high-dimensional feature matrices. Furthermore, Bento is a member of the Scverse ecosystem [32], enabling integration with Scanpy [21], Squidpy [17], and more than 30 other single-cell omics analysis tools.

## Results

### Overview of Bento data infrastructure for subcellular analysis

In order to facilitate a flexible workflow, Bento is generally compatible with molecule-level resolution spatial transcriptomics data (Fig. 1A), such as datasets produced by MERFISH [13], seqFISH + [14], CosMx (NanoString) [33], Xenium (10 × Genomics) [15, 34], and Molecular Cartography (Resolve Biosciences) [35]. Bento's workflow takes as input (1) 2D spatial coordinates of transcripts annotated by gene and (2) segmentation boundaries (e.g., cell membrane, nuclear membrane, and any other regions of interest) (Fig. 1B). While 3D molecular coordinates are commonly included, 3D segmentation information is limited to z-stacked 2D segmentation, limiting its usability. If available, Bento can also handle arbitrary sets of segmentations for other subcellular structures or regions of interest. These inputs are stored in the AnnData data format [36], which links cell and gene metadata to standard count matrices, providing compatibility with standard single-cell RNA-seq quality control and analysis tools in the Scverse ecosystem [32]. With a data structure for segmentation boundaries and transcript coordinates in place, Bento can easily compute spatial statistics and measure spatial phenotypes to build flexible multidimensional feature sets for exploratory subcellular analysis and utilize these spatial metrics to augment quality control (Fig. 1C).

Bento offers a precise yet flexible palette of novel complementary subcellular analyses (Fig. 1D). We introduce RNAforest, a multilabel approach for annotating RNA

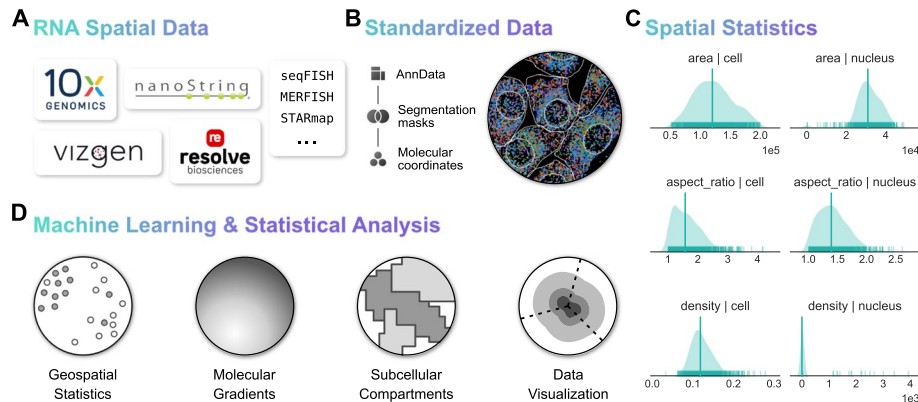

**Fig. 1** Workflow and functionality of the Bento toolkit. **A** Single-molecule resolved spatial transcriptomics data from commercial or custom platforms are ingested into Bento where it is converted to the AnnData format (**B**), where it can be manipulated with Bento as well as a wide ecosystem of single-cell omics tools. **C** Geometric statistics are illustrated for the seqFISH + dataset, including metrics describing cell and nuclear geometries and cell density to assess overall data quality. **D** Bento has a standard interface to perform a wide variety of subcellular analyses

localization patterns adapted from FISH-quant v2 [22]. We find that many RNAs are spatially distributed according to gene function. We then implement RNAcoloc, a context-specific approach to quantify colocalization to characterize how genes colocalize with each other in a compartment-specific manner. Having established systematic patterning and organization of RNA transcripts, we demonstrate RNAflux, an unsupervised method for semantic segmentation of subcellular domains. RNAflux first quantifies subcellular expression gradients at pixel resolution before identifying consistent subcellular domains via unsupervised clustering. We demonstrate the utility of Bento's tools by applying them to identifying critical localization changes in human iPSC-derived cardiomyocytes upon drug treatment with doxorubicin, a widely used chemotherapeutic known to cause cardiotoxicity [37].

### RNAforest: utilizing subcellular landmarks to predict RNA subcellular localization

In computer vision, key points or landmarks are commonly used for tasks like facial recognition [38] and object detection. Analogous to these classical applications, we derive spatial features using cell and nucleus boundaries as landmarks to predict RNA localization patterns from spatial summary statistics. Building on the summary statistics used for classifying smFISH data in FISH-quant v2 [39], RNAforest consists of an ensemble of five binary random forest classifiers rather than a single multi-classifier model to assign one or more labels. These pattern labels, adapted from several high-throughput smFISH imaging experiments in HeLa cells [40–43], are broadly applicable to eukaryotic cells: (i) nuclear (contained in the volume of the nucleus), (ii) cytoplasmic (diffuse throughout the cytoplasm), (iii) nuclear edge (near the inner/outer nuclear membrane), (iv) cell edge (near the cell membrane), and (v) none (complete spatial randomness). It is important to note, as was done previously in FISH-quant v2 [39] that because of the 2D nature of the dataset, RNA that is in truth cytoplasmic but above or below the nucleus will still appear as though in the nucleus when collapsed in the *z*-dimension. As we use the FISH-quant v2 pattern simulation framework, this is accounted for in the training dataset.

We used the FISH-quant v2 simulation framework to generate realistic ground-truth data [42]. Each sample is defined as a set of points with coordinates in two dimensions, representing the set of observed transcripts for a gene in a particular cell. In total, we simulated 2000 samples per class for a total of 10,000 samples (see the " Methods" section). We used 80% of the simulated data for training and held out the remaining 20% for testing (Additional File 1: Fig. S1A). Each sample is encoded by a set of 13 input features, describing characteristics of its spatial point distribution, including proximity to cellular compartments and extensions (features 1–3), measures of symmetry about a center of mass (features 4–6), and measures of dispersion and point density (features 7–13) (Fig. 2A). These features are normalized to morphological properties of the cell to control for variability in cell shape. A detailed description of every feature is described in Additional File 1: Table S1, and model architectures and hyperparameters are described in Additional File 2: Table S2 (see the " Methods" section).

We applied RNAforest on two datasets from different spatial platforms, cell types, and gene panel sizes: a MERFISH dataset in U2-OS cells and a seqFISH+dataset in 3T3

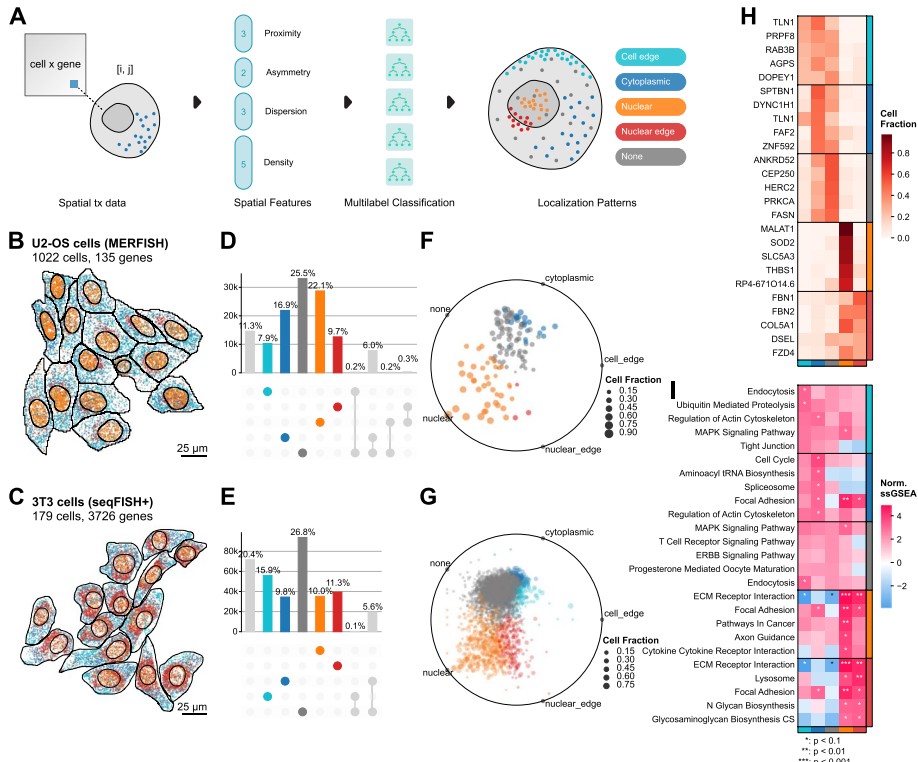

**Fig. 2** Subcellular localization pattern identification with RNAforest. **A** Thirteen spatial summary statistics are computed for every gene-cell pair describing the spatial arrangement of molecules and boundaries in relation to one another. The features (Supp. Table 1) are inputs for RNAforest, a multilabel ensemble classifier that assigns one or more subcellular localization labels: cell edge, cytoplasmic, nuclear, nuclear edge, and none. The colors for each label are used consistently throughout to figure. Top 10 genes for each label visualized for each label other than "none" in **B** U2-OS cells and **C** 3T3 cells. **D** and **E** are UpSet plots showing the proportion of measured transcripts assigned to each label. **F** and **G** show the relative label proportion across cells for each gene and are colored by the majority label (**F** and **G**). **H** Top 5 consistent genes for each label. **I** ssGEA identifies the enrichment of GO cellular component domains for each label in the 3T3 cell dataset. Stars denote *p*-values under thresholds defined in the legend. *P*-values are derived from ssGSEA permutation tests with Benjamini–Hochberg correction controlling for false discovery rate

cells. Validation performance on manual annotation of subsets of both datasets shows that RNAforest generalizes well despite biological and technical differences (see the " Methods" section, Additional File 1: Fig. S1B). The MERFISH dataset measured 130 genes (low plexity) with high detection efficiency per gene (111 molecules per gene per cell on average), while the seqFISH + dataset measured 10,000 genes (very high plexity) with lower detection efficiency (8 molecules per gene per cell on average) (Fig. 2B, C, Additional File 1: Fig. S1C-F). In agreement with previous work characterizing RNA localization of 411 genes [43], we find that genes commonly exhibit variability in localization across cells. This suggests that heterogeneity in localization likely generalizes to the entire transcriptome. Of the localization patterns besides "none," "nuclear" was the most common (22.1%) in the U2-OS osteosarcoma cells (Fig. 2D, F), while "cell edge" was the most common (15.9%) in the 3T3 fibroblast cells (Fig. 2E, G).

In the U2-OS cells, we found many genes to have preferential localization in different subcellular compartments (Fig. 2H). In agreement with our RNAflux findings, we find genes known to localize to the nucleus [44, 45] to be frequently labeled "nucleus" (MALAT1, SOD2) and genes encoding secreted extracellular proteins [13] to be frequently labeled "nuclear edge" (FBN1, FBN2). As expected, we find genes preferentially "nuclear" and "nuclear edge" localized to mirror nucleus and endoplasmic reticulum genes found in a 10 k genes MERFISH study of U2-OS cells that included ER staining [46] (Additional File 1: Fig. S2; see the " Methods" section). Leveraging the 3T3 seqFISH + dataset's higher plexity, we were able to ask whether genes with similar localization preferences are functionally related. We applied gene set enrichment analysis to gene localization frequencies to identify enriched gene ontology terms [47] (Fig. 2I; see the " Methods" section). Secretory processes were enriched in the nucleus and nuclear edge, which may be linked to increased transcription of fibroblast-related functions. Cell edge enriched pathways consisted of those with the cell membrane as their site of function (e.g., endocytosis and tight junction suggesting local translation of these genes). Additionally, the term for cell cycle was significantly enriched in the cytoplasm only. Genes without strong localization preference (most frequently "none") were not significantly associated with any pathways. These genes likely do not undergo active transport and are functionally independent of local translation.

In summary, RNAforest gives a user a facile method for annotating RNA localization patterns and quantifying heterogeneity in a transcriptome-wide manner independent of RNA abundance. Beyond known RNA localizations, we find that transcript location is generally associated with known gene function, alluding to the systematic spatial regulation of RNA transport. We foresee RNAforest will be a valuable addition to characterize RNA localization across diverse spatial transcriptomics datasets.

### RNAcoloc: an approach for context-specific RNA colocalization

In geospatial information processing, a fundamental feature that is often gleaned from large datasets is the colocation of objects (e.g., gleaning socialization metrics from cell phone colocation data in Singapore [48]). Colocation is similarly valuable in understanding co-translation and interaction networks of genes in a biological context [49]. Recent spatial transcriptomics approaches have used a number of colocalization metrics from the geographic information systems and ecology fields, e.g., the bivariate

versions of Ripley's K function (also known as cross-k-function) [50], Moran's *I* [51], and the join count statistic [52]. These metrics are designed to measure spatial associations between two populations, i.e., gene A transcripts and gene B transcripts. However, it is more appropriate to think of all transcripts in a single cell from a single population; after all, RNA transcription and localization are not completely stochastic. We have shown that the subcellular distribution of RNA is highly structured with RNAforest. As such, we developed RNAcoloc, an approach that combines the Colocation Quotient (CLQ) [53] metric and tensor decomposition for context-specific RNA colocalization (see the "Methods" section). The CLQ is a colocalization statistic that is capable of accounting for the biophysical properties of RNA spatial distributions. First, the CLQ considers how clustered the overall RNA population is in a cell and measures whether specific pairs of genes are more clustered than expected given the spatial pattern of the overall population. Second, the CLQ is inherently asymmetric and captures the direction of attraction, i.e., the attraction of gene A to gene B is not the same as the attraction of gene B to gene A. This is most common when gene A and gene B have very different expression levels, which is prevalent due to overdispersion in gene expression data.

RNAcoloc calculates CLQ scores for each gene per cell in a compartment-specific manner, such that each sample has 2 scores, a nucleus and cytoplasm CLQ score. An initial comparison in the U2-OS dataset of global colocalization between nuclear and cytoplasmic fractions unsurprisingly found that transcripts from the same gene tend to cluster more tightly with themselves than with transcripts from other genes (Fig. 3B). Additionally, self-colocalization is significantly stronger in the cytoplasm than in the nucleus. In conjunction with our findings from RNAforest analysis that genes of the same localization pattern tend to have similar functions, this suggests that the RNAs are more tightly spatially regulated once exported from the nucleus.

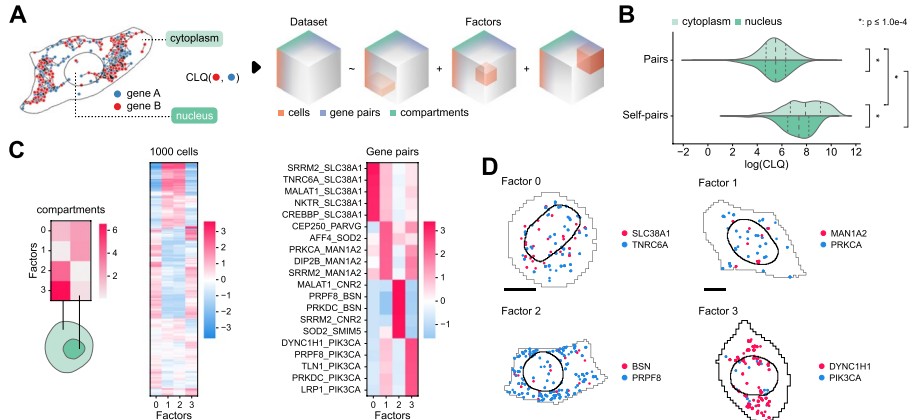

**Fig. 3** Compartment-specific RNA colocalization with RNAcoloc. **A** Transcripts are separated by compartment (nucleus and cytoplasm) before CLQ scores are calculated for every gene pair across all cells. This yields a cell × gene pair × compartment tensor. **B** Pairwise comparison of log CLQ distributions for gene pairs and self-pairs, further categorized by compartment. The Mann–Whitney *U* test was used for comparisons. Stars denote *p*-values below the legend threshold with Benjamini–Hochberg correction controlling for false-discovery rate. From top to bottom, group sizes are 12,254,430 (cytoplasm gene pairs), 115,187 (nucleus gene pairs), 6,778,402 (cytoplasm self-pairs), and 86,474 (nucleus self-pairs). **C** Tensor decomposition yields 4 factors. From left to right, the three heatmaps show the loadings of each factor for each dimension—compartments, cells, and gene pairs. Only the top 5 associated gene pairs for each factor are shown. **D** Top examples of compartment-specific colocalized gene pairs. Black scale bars denote 10 μm

By calculating CLQ scores for every gene–gene pair across compartments, RNAcoloc constructs a tensor of shape $P \times C \times S$ where $P$, $C$, and $S$ represent the number of gene–gene pairs, cells, and compartments, respectively (Fig. 3A; see the " Methods" section).

RNAcoloc then applies tensor decomposition — specifically, non-negative parallel factor analysis — a data-driven, unsupervised approach for discovering substructure in high-dimensional data [31, 54] to decompose the tensor into $k$ "factors". The number of factors is determined using the elbow method heuristic, optimizing for the root mean squared error (RMSE) reconstruction loss (see the " Methods" section). Unlike matrix dimensionality reduction methods, such as PCA, the order of the components (factors) is unassociated with the amount of variance explained. Each factor is composed of 3 loading vectors, which correspond to the compartments, cells, and gene pairs. Higher values denote a stronger association with that factor. Crucially for interpretation, factors derived from tensor decomposition are not mutually exclusive and can share overlapping sets of associated compartments, cells, and gene pairs.

Applied to the U2-OS dataset, RNAcoloc decomposes RNA colocalization into 4 factors. Examining factor loadings indicates two distinct subpopulations of cells with compartment-specific colocalization behaviors; cluster 1 cells exhibit uniform (Factor 0) and cytoplasmic (Factor 3) colocalization, while cluster 2 cells show nuclear (Factor 1) and cytoplasmic colocalization (Factor 2) (Fig. 3C, D). Factor 3 describes the colocalization of gene pairs in the cytoplasm of cluster 1 cells, especially a number of genes that attract PIK3CA transcripts While little is known about PIK3CA RNA interactions, the PI3K pathway regulates mitotic organization, including the regulation of dynein and dynactin motor proteins. DYNC1H1 is among the top genes attracting PI3KCA and specifically encodes cytoplasmic dynein, a motor protein critical for spindle formation and chromosomal segregation in mitosis [55]. This hints that not only is compartmental localization of RNA linked to the cell cycle [46], but RNA-RNA interactions may play a role as well. In cluster 2 cells, MALAT1 attracts CNR2 transcripts more than expected in the cytoplasm. Even though MALAT1 is canonically abundantly localized to the nucleus, this demonstrates that the CLQ score identifies gene pairs colocalizing more than expected despite the disproportionate expression of MALAT1 relative to CNR2, whereas other approaches seem confounded by large differences in expression [45].

We demonstrate the ability of RNAcoloc to quantify compartment-specific gene-pair colocation by exploring cytoplasmic vs. nuclear colocalization. As we found separately with RNAforest, RNAcoloc analysis finds evidence that RNA transport is spatially regulated, especially after nuclear export. We highlight several examples of colocalization suggesting how RNA localization allows the same gene to have multiple functions in a spatially dependent fashion, i.e., depending on its molecular neighbors and local environment [56, 57]. We foresee RNAcoloc will be increasingly relevant as many spatial technologies are beginning to image proteins along with RNA, which can be used to delineate more granular compartments, such as cell organelles or distinct regions, e.g., neuron cell bodies vs dendrites.

### RNAflux: unsupervised semantic segmentation of subcellular domains in single cells

To build on RNAforest, we overcame the restricted number of localization patterns defined by the supervised method by framing RNA localization as an unsupervised

embedding problem. RNAflux looks at local neighborhoods within the space of a cell and builds a normalized gene composition per neighborhood. Differences in neighborhood compositions can be leveraged to identify distinct subcellular domains in a manner that is entirely unsupervised and independent of cell geometry.

We applied this embedding procedure to compute a gene composition vector for every pixel in 2D coordinate space, generating a spatially coherent embedding across entire cells (Fig. 4A; see the " Methods" section).

Applied to a MERFISH dataset with a target panel of 130 genes across over 1153 U2-OS cells, we demonstrate that RNAflux embeddings can detect transcriptionally distinct subcellular domains. Performing dimensional reduction of the embeddings showed that the top sources of variation spatially correspond to the nucleus, the nuclear periphery, and cytoplasmic regions consistently across cells (Fig. 4B; see the " Methods" section) confirming that RNAflux measures intracellular transcriptional variation, as opposed to intercellular variation. To delineate compositionally similar domains in a

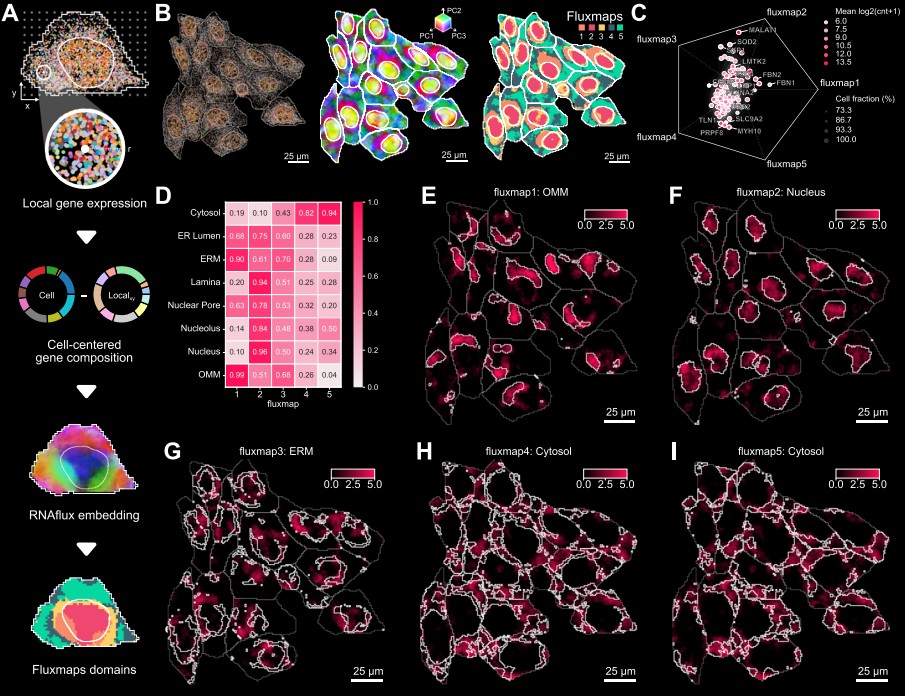

**Fig. 4** RNAflux finds distinct subcellular domains with consistent spatial organization and local gene composition. **A** Flowchart of RNAflux and fluxmap computation. Local neighborhoods of a fixed radius are arrayed across a cell and a normalized gene composition is computed for each pixel coordinate, producing an RNAflux embedding. The first three principal components of the RNAflux embedding are visualized for U2-OS cells coloring RGB values by PC1, PC2, and PC3 values respectively for each pixel. Fluxmap domains are computed from each RNAflux embedding to create semantic segmentation masks of each subcellular domain. **B** The left panel shows a field of view of U2-OS cells, dots denoting individual molecules colored by gene species, nuclei, and cell boundaries outlined in white. For the same field of view of cells, the center panel shows RNAflux embeddings and the right panel shows fluxmap domains. **C** The scatter plot shows how the composition of each gene is distributed across fluxmap domains. The position of each point denotes the relative bias of a given gene's composition across fluxmaps. **D** Heatmap showing the fraction of pixels with a positive enrichment value for each APEX-seq location for each fluxmap domain. **E–I** The most highly enriched location is shown for each fluxmap domain. Domain boundaries are denoted by white lines within each cell

data-driven manner, we cluster pixel embeddings using self-organizing maps (SOMs), effectively performing unsupervised semantic segmentation (see the " Methods" section). We denote the resulting clusters as "fluxmap domains." We found that this assigned pixels to 5 fluxmap domains, consistently highlighting spatial regions across every cell (e.g., fluxmap 2 is always nuclear while the remaining domains constitute the cytoplasm) (Fig. 4B). By considering the spatial distribution of molecules across fluxmap domains, we can quantify the composition of molecules for each gene across fluxmaps (Fig. 4C), e.g., nuclear-localized MALAT1 [44].

We sought to characterize the fluxmap domains with known information about RNA localization. We used data from a previous study that measured gene expression at "distinct subcellular locales" via APEX-seq, a technique for proximity labeling and sequencing of RNA [58]. Of the 3288 genes differentially enriched to one or more locales, 63 overlapped with the 130 MERFISH genes. The location enrichment score for each pixel is calculated by taking the weighted sum of its RNAflux embedding and the measured relative enrichment, i.e., log fold change measured by APEX-seq loadings for a given organelle-specific geneset (see the " Methods" section). Visualizing each pixel's location-specific enrichment scores from the APEX-seq dataset highlights the subcellular localization of these compartments, including the cytosol, nucleus, nucleolus, nuclear pore, nuclear lamina, endoplasmic reticulum lumen (ER lumen), ER membrane (ERM), and the outer mitochondrial membrane (OMM) (Fig. 4D). We find the nuclear compartments have high scores in domain 2, while the cytoplasm scores rank highest in domains 4 and 5. Both the ERM and OMM scores are the strongest in domain 1 (Fig. 4E).

The most common application for spatial transcriptomics is in mapping heterogeneous cell types in large tissue samples. This presents several challenges. First, the panels for these experiments are weighted heavily towards cell type markers determined by single-nuclei RNA-seq, potentially reducing the intracellular variability in the expression on which Bento relies. Second, substantial intercellular heterogeneity can skew the low-rank embedding by introducing too much variance in requisite cell radii information. To explore the applicability of RNAflux on tissue, we applied it to a previously published breast cancer tissue dataset generated by $10 \times$ Xenium [59]. We successfully reproduced the identification of unique cell types that reflect canonical expression markers (Fig. S4A and B). We apply RNAflux to cell type-disaggregated subsets of the full datasets in fields of view of interest with at least 100 cells of interest and find that fluxmaps 1–3 show different enrichment scores for Nucleus and Endoplasmic Reticulum (ER; combination of ERM and ER lumen). We note that despite the successful delineation of discrete regions in the form of fluxmaps when looking at different regions of tissue that are enriched for different cell types, nuclear and ER enrichment scores change for each fluxmap (Fig. S4C).

In summary, RNAflux finds distinct subcellular domains with consistent spatial organization and local gene composition. As an unsupervised method, RNAflux can be applied to any cell type for inferring subcellular domains from transcript locations and functionally annotated with biological enrichment analysis. This process is best performed on cell type-separated data to guard low-rank embeddings from being generated by cells of vastly different morphologies. This disaggregation is important because, unlike uniform cell lines, heterogeneous tissue composed of functionally diverse stromal

cells and leukocytes should indeed be expected to have different distributions and complements of subcellular domains.

### Doxorubicin-induced stress in cardiomyocytes depletes RNA from the endoplasmic reticulum

Having established Bento's utility to characterize RNA localization in U2OS cells, we applied Bento to quantify changes in localization in the context of perturbations in cells. Specifically, we performed single-molecule spatial transcriptomics on doxorubicin-treated and untreated cardiomyocytes to identify RNA localization changes as a result of treatment.

Doxorubicin (DOX) was once one of the most effective broad-spectrum anti-cancer anthracycline antibiotics [60, 61] with particular efficacy against solid malignancies such as lung and breast cancer, as well as hematologic neoplasia [62, 63]. However, DOX's propensity to cause cardiac damage in patients has led to significant limitations in its clinical use [64]. The exact mechanism by which DOX induces heart failure is unclear, but significant evidence suggests cardiomyocyte injury driven by oxidative stress is a major factor [62, 65–68]. Specifically, DOX causes stress and dysfunction in multiple cellular compartments in cardiomyocytes such as mitochondria, Sarco/endoplasmic reticulum (SER), deficiencies in calcium signaling, and lipid degradation at the cellular membrane [69]. We reasoned that by measuring the localization of the RNA transcripts of 100 genes crucial to cardiomyocyte health and function (Additional File 2: Table S2) and leveraging the tools developed within Bento, we could recapitulate known dysfunction of subcellular domains in cardiomyocytes upon DOX stress and measure novel RNA localization phenotypes.

We utilized a chemically defined protocol to differentiate human induced pluripotent stem cells (iPSCs) into beating cardiomyocytes and treated them with either DMSO (vehicle) or 2.5 µM DOX for 12 h before fixation (see the " Methods" section). Single-molecule spatial transcriptomes were measured by Resolve Bioscience using Molecular Cartography. The resulting data was segmented using ClusterMap [24] for cell boundaries and Cellpose [70] for nuclei boundaries (Fig. 5A). Non-myocytes were filtered out using SLC8A1 as a canonical marker for cardiomyocytes (see the " Methods" section, Additional File 1: Fig. S3A). Comparing vehicle and DOX-treated cardiomyocytes, we found NPPA, a classic marker for cardiac stress [71, 72], to be upregulated in DOX-treated cells (Fig. 5B). We identified subcellular domains in the vehicle and DOX-treated cardiomyocytes using RNAflux, clustering the domains into four fluxmap domains (Fig. 5C, D, Additional File 1: Fig. S3B). Enrichment of location-specific gene expression aligned domains to the nucleus (nuclear pore, nucleolus, and nucleus), ERM and OMM, ER lumen, and cytosol respectively (Fig. 5E, Additional File 1: Fig. S3C). Comparing the gene composition in each domain, we observe an overall localization bias towards both the nucleus and ERM/OMM in vehicle-treated cells (Fig. 5E top), in agreement with prior poly(A) smFISH studies [73]. However, RNA in the DOX-treated cardiomyocytes demonstrated a shift in average RNA localization away from the ERM/OMM and towards the nucleus (Fig. 5E bottom). There was no correlation when comparing the logFC in expression and the difference in nuclear composition of genes after treatment, indicating that localization towards the

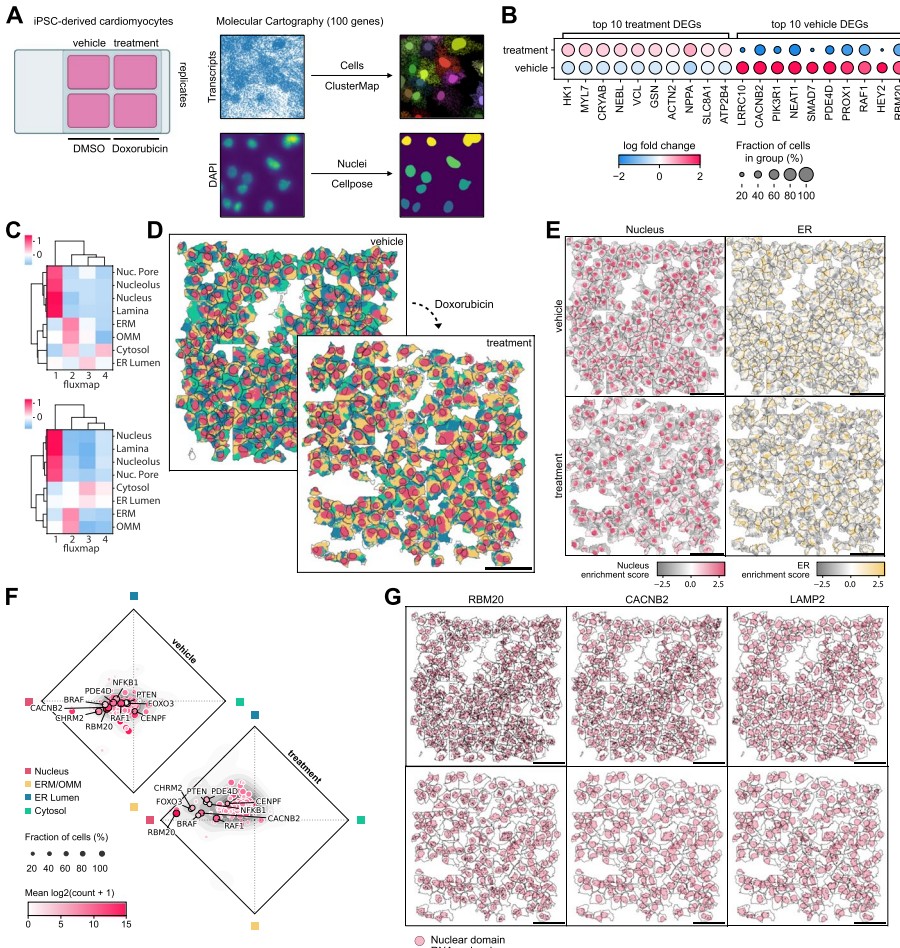

**Fig. 5** Subcellular RNA localization changes upon Doxorubicin treatment in iPSC-derived cardiomyocytes. **A** Cardiomyocytes derived from human iPSCs were treated with DMSO or 2.5 μM DOX for 12 h. The localizations of 100 genes relevant to cardiomyocyte health and function were measured using Molecular Cartography. Cell boundaries were determined using ClusterMap and nuclei were segmented using Cellpose. **B** Top 10 differentially upregulated and downregulated genes in vehicle versus treatment. *T*-test was used for comparisons. All genes shown are significant given an adjusted *p*-value threshold of *p* < 0.01. Benjamini–Hochberg correction was used to control for the false discovery rate. Vehicle and treatment conditions have *n* = 7159 and 6260 cells respectively. **C** APEX-seq location-specific gene enrichment of fluxmap domains for the cytosol, endoplasmic reticulum membrane (ERM), endoplasmic reticulum lumen (ER Lumen), nuclear lamina, nucleus, nucleolus, nuclear pore, and outer mitochondrial matrix (OMM). **D** Fluxmap domains visualized for a representative field of view of cardiomyocytes for vehicle and treatment respectively highlighting cellular nuclei, ERM/OMM, ER Lumen, and cytosol. **E** RNAflux fluxmap enrichment of each gene averaged across vehicle and treatment cardiomyocytes captures changes in subcellular RNA localization. Top 10 genes are labeled and ranked by the largest shifts between compartment compositions. Shifts are quantified by Wasserstein distance. **F** Average gene enrichment in each fluxmap across vehicle and treatment conditions colored by log-fold expression demonstrates population-level shifts in transcript subcellular localization. **G** Visualization of RBM20, CACNB2, and LAMP2 transcripts confirms the depletion of transcripts from the perinuclear and cytosolic compartments of cardiomyocytes upon DOX treatment

nucleus is not driven by transcript abundance (Spearman *r* = 0.07, *p* = 0.4944). There is evidence that 90% of genes have a half-life of less than 260 min [74], far less than the 12-h DOX treatment, indicating that the shift in RNA localization is likely due to reduced nuclear export of newly synthesized RNA from the nucleus to the ERM/

OMM. Indeed, even low concentrations of DOX have been demonstrated to alter structural fibrous proteins as well as mitochondrial depolarization and fragmentation [75]. Of particular note, the RNA binding protein RBM20—a critical regulator of mRNA splicing of genes encoding key structural proteins associated with cardiac development and function—had a pronounced depletion of RNA transcripts outside of the nucleus upon DOX treatment (Fig. 5G left). With further validation, this may indicate nuclear retention and or degradation of nuclear exported RBM20 mRNA as a potential mechanism of DOX-induced cardiomyopathy. Similarly, we found the mRNA of calcium voltage-gated channel subunit CACNB2 to also deplete outside of the nucleus (Fig. 5G middle). The loss of CACNB2 translation outside of the nucleus may impact calcium signaling crucial to cardiomyocyte function [76]. Most genes only showed weak shifts in localization, similar to LAMP2 (Fig. 5G right).

## Discussion

Bento seeks to interrogate biology via its "subcellular first" approach to spatial analysis, complementary to "cell-type or tissue first" spatial analysis methods. The toolkit enables quantitative, reproducible, and accessible analysis agnostic to spatial technology platforms in a standardized framework. We implement three novel methods to interrogate subcellular RNA organization: RNAforest for supervised annotation of localization patterns, RNAcoloc for compartment-aware colocalization analysis, and RNAflux for identifying transcriptionally distinct subcellular domains. We showed that with RNAflux, we were able to quantify RNA localization in a variety of contexts, including domain-specific gene localization, drug-induced changes in localization, and cell type-specific localization. With both RNAflux and RNAforest, we find that subcellular mRNA localization reflects gene function. With RNAcoloc, we explore the use of CLQ scores to quantify pairwise gene colocalization in the context of asymmetric associations.

From these results, we found three main factors to limit the effectiveness of subcellular-resolution analysis: molecule density, segmentation quality, and target panel composition. In particular, RNAflux becomes uninformative if too few molecules are detected per cell or if the number of molecules per gene is too sparse. We found that datasets with higher density, i.e., molecules per micrometer$^2$, are less noisy and inform more coherent gradients and domains. Both the U2-OS dataset and cardiomyocyte datasets had high enough molecule density to identify consistent fluxmaps. Notably, RNAflux robustly highlights domains corresponding to the nucleus and ER despite some poor cell segmentation and partially unannotated nuclei in the cardiomyocytes dataset (Fig. 5D). As most commercial target panels are largely composed of marker genes for cell type identification, RNAflux embeddings should be interpreted carefully, especially if transcripts show little spatial variation in subcellular localization. In contrast, RNAforest performs reliably beyond a minimum of 5–10 molecules per sample but is sensitive to accurate segmentation for calculating cell morphology-dependent features (Additional File 1: Fig. SE-F). The 3T3 cells were manually segmented and the U2-OS cells had relatively accurate segmentation and were therefore amenable to applying RNAforest. We found that the segmentation in the cardiomyocytes is accurate enough for single-cell gene expression analysis, but lacked the precision needed to apply RNAforest. In the case of RNA-coloc, the limiting factor to identifying relevant biology is the target panel composition.

The current focus of most target panels typically includes cell type markers and highly expressed genes, whereas it would be more informative to identify colocalizing members of protein complexes, functional pathways, or ligand-receptor pairs. Our ability to characterize relevant pathways with our curated cardiomyocyte gene panel shows how gene panel design focused on function enables discovery. Ultimately, non-targeted transcriptome-scale technologies will be necessary to unlock the full potential of subcellular biology.

A dimensional limitation of Bento is its current inability to process three-dimensional spatial transcriptomic data. While some commercially available spatial transcriptomic methods yield RNA molecular coordinates in 3D, the nuclear and cell segmentation is inevitably still two-dimensional making it difficult to interpret z-dimensional positions lacking the context of cellular geometry in 3D. However, the algorithms behind RNA-forest, RNAcoloc, RNAflux, and the plethora of feature calculation functions in Bento are inherently extensible to leveraging three-dimensionality. When three-dimensional cell segmentation improves, we intend to extend Bento to support three-dimensional analysis.

## Conclusions

Conventionally, RNA is treated as an intermediary vehicle encoding genomic information for protein synthesis. We began our investigation of RNA localization with the hope of understanding how the spatial organization of RNA functions as a mechanism for post-transcriptional regulation. However, RNAflux conceptually introduces using RNA molecular coordinates as a latent layer of information encoding cellular space–time. Here, we used that latent layer of information to identify subcellular domains. As spatial omic technologies improve to capture more and more information, the potential applications of such latent embeddings will grow as well. Indeed at the tissue level, this concept is already being leveraged with a recent tool, TensionMap, using RNA localization information to predict mechanical tension [77]. As applications for spatial transcriptomics grow in popularity and complexity, we envision that Bento is a platform for the next generation of tools needed to quantify the complex molecular dynamics governing normal and abnormal cellular processes.

## Methods

### MERFISH and seqFISH + data preprocessing

For the seqFISH + dataset, we limited the scope of our analysis to the set of genes for which at least 10 molecules were detected in at least one cell. This helped reduce sparsity in the data, resulting in 3726 genes remaining. Because pattern classification requires nuclear segmentation masks, we removed all cells lacking annotated nuclei for the remainder of 179 cells. Because the MERFISH data had a much higher number of molecules detected per gene, no gene needed to be removed. Again, cells without annotated nuclei were removed, leaving 1022 cells for downstream analysis.

### Preprocessing cardiomyocytes datasets

Single-cell expression matrices of both vehicle replicates and both DOX treatment samples were concatenated as a single expression matrix. Cells were projected into

two dimensions with UMAP dimensional reduction. No significant batch effects were detected. Leiden clustering was performed at resolution=0.5 to isolate and filter out a non-myocyte population depleted in SLC8A1 expression (Additional File 1: Fig. S3A). All described preprocessing steps were performed in Scanpy [21].

### RNAforest: model selection and training

We evaluated 4 base models for the multilabel classifier including random forests (RF), support vector machines (SVM), feed-forward fully-connected neural networks (NN), and convolutional neural networks (CNN). While all other models use the 13 spatial features for input (Additional File 1: Table S1), the CNN takes $64 \times 64$ image representations of each sample as input. Each multilabel classifier consists of 5 binary classifiers with the same base model. We used the labeled 10,000 simulated samples for training, stratifying 80% of the simulated data for training and holding out the remaining 20% for testing. To select the best hyperparameters for each multilabel classifier, we sampled from a fixed hyperparameter space with the Tree-structured Parzen Estimator algorithm and evaluated performance with fivefold cross-validation (Additional File 2: Table S2). We retrained the final model (random forest base model) on all training data with the best-performing set of hyperparameters (Additional File 1: Fig. S1E). Exact steps can be found and reproduced in notebooks stored in the GitHub repository: https://github.com/ckmah/bento-manuscript.

### RNAforest: image rasterization of molecules and segmentation masks for CNN

To generate an image for a given sample, point coordinates, the cell segmentation mask, and the nuclear segmentation mask are used. The area of the cell is tiled as a $64 \times 64$ grid, where each bin corresponds to a pixel in the final image. Values are stored in a single channel to render a grayscale image. Pixels inside the cell are encoded as 20 and inside the nucleus encoded as 40. Bins with molecules are encoded as $(40 + 20 \times n)$ where $n$ is the number of molecules. Finally, values are divided by 255 and capped to be between 0 and 1.

### RNAforest: simulating training data

We trained a multilabel classifier to assign each gene in every cell labels from five categories: (i) nuclear (contained in the volume of the nucleus), (ii) cytoplasmic (diffuse throughout the cytoplasm), (iii) nuclear edge (near the inner/outer nuclear membrane), (iv) cell edge (near the cell membrane), and (v) none (complete spatial randomness). These categories are a consolidation of those observed in several high-throughput smFISH imaging experiments in HeLa cells [40–43]. We used the FISH-quant simulation framework (https://code.google.com/archive/p/fish-quant/) to generate realistic ground-truth images using empirically derived parameters from the mentioned high-throughput smFISH imaging experiments in HeLa cells [42]. In total, we simulate 2000 samples per class for a total of 10,000 training samples.

1. *Cell shape*: Cell morphology varies widely across cell types, and for classifier generalizability, it is important to include many different morphologies in the training set. We use a catalog of cell shapes for over 300 cells from smFISH images in HeLa cells

that capture nucleus and cell membrane shape [42]. Cell shapes were obtained by cell segmentation with CellMask and nuclear segmentation was obtained from DAPI staining.

2. mRNA *abundance*: We simulated mRNA abundance at three different expression levels (40, 100, and 200 mRNA per average-sized cell) with a Poisson noise term. Consequently, total mRNA abundance per cell was between 5 and 300 transcripts.

3. Localization *pattern*: We focused on 5 possible 2D localization patterns, including cell edge, cytoplasmic, none, nuclear, and nuclear edge. Each pattern was further evaluated at 3 different degrees—weak, moderate, and strong. Moderate corresponds to a pattern typically observed in a cell, whereas weak is close to spatially random. These 5 classes aim to capture biologically relevant behavior generalizable to most cell types; there is room for additional classes describing other biologically relevant localization patterns so long as they can be accurately modeled.

### RNAforest: manual annotation of true biological validation data

Using 3 individual annotators, we annotated the same 600 samples across both datasets, keeping samples with 2 or more annotator agreements as true annotations, resulting in 165 annotated seqFISH + samples and 238 annotated MERFISH samples (403 total). We used Cohen's kappa coefficient [78] to calculate agreement between pairs of annotators for each label yielding an overall coefficient of 0.602. We found that pairwise agreement between annotators across labels was fairly consistent ranging between 0.588 and 0.628, while label-specific agreement varied more, ranging between 0.45 and 0.72 (Additional File 4: Table S4).

### RNAforest: functional enrichment of gene pattern distributions

For enrichment of compartment-specific expression from Xia et al. [46], scores are calculated by taking the weighted sum of gene pattern frequencies and published compartment log fold-change values (Additional File 1: Fig. S2). The Benjamini−Hochberg correction was used to correct *p*-values for multiple hypothesis testing.

For the seqFISH + dataset, we performed a single-sample Gene Set Enrichment Analysis [79, 80] on gene pattern frequencies to compute enrichment scores (Fig. 3I). ssGSEA was performed with the GSEApy Python package and the "GO_Cellular_Component_2021" gene set library curated by Enrichr [81]. Gene sets with a minimum size of 50 and a maximum size of 500 were analyzed.

### Colocation quotient for RNA colocalization analysis

Pairwise colocalization of genes was determined for each compartment of every cell separately. In this case, each cell was divided into compartments, cytoplasm, and nucleus. The colocation quotient (CLQ) was calculated for every pair of genes *A* and *B*. The CLQ is defined as an odds ratio of the observed to expected proportion of *B* transcripts among neighbors of *A* for a fixed radius *r*; it is formulated as:

$$CLQ_{A \to B} = \frac{C_{A \to B}/N_A}{N\prime_B/(N-1)}$$

Here $C_{A \to B}$ denotes the number of $A$ transcripts of which $B$ transcripts are considered a neighbor. $N_A$ denotes the total number of $A$ transcripts, while $N'_B$ stands for the total number of $B$ transcripts. In the case that $A = B$, $N'_B$ equals the total number of $B$ transcripts minus 1. $N$ denotes the total number of transcripts in the cell. Following statistical recommendations from the original formulation of the colocation quotient (CLQ), genes with fewer than 10 transcripts were not considered to reduce sparsity and improve testing power [53].

### Tensor decomposition for compartment-specific colocalization

We begin by calculating the CLQ for every pair of genes within each compartment of every cell. We structure our data as follows:

1. Cells set: Denote the set of cells as $C = \{c_1, c_2, ..., c_n\}$, where $c_i$ represents the $i^{th}$ cell and $n$ is the total number of cells.
2. Compartments set: Every cell has the same set of compartments, represented as $K = \{k_1, k_2, ..., k_m\}$, where $k_j$ is the $j^{th}$ compartment within a cell.
3. Gene pairs set: The gene pairs are represented by $G = \{g_1, g_2, ..., g_p\}$ where $g_p$ is the $p^{th}$ gene pair.

By computing the CLQ for every combination of cells in $C$, compartments in $K$, and gene pairs in $G$, we populate a three-dimensional tensor $X$ with dimensions corresponding to these sets.

We then apply non-negative parallel factor analysis (PARAFAC) as implemented in Tensorly [31] to reduce the dimensionality of our dataset and capture the underlying patterns. For tensor decomposition, we employed non-negative parallel factor analysis as implemented in Tensorly [31]. The tensor $X$ is decomposed into the sum of $R$ factors; each factor is a three-dimensional tensor expressed as the outer product of three vectors: $x_r$ (compartment loadings), $y_r$ (cell loadings), and $z_r$ (gene pair loadings). This is denoted as follows:

$$\widehat{X}_R = \sum_{r=1}^{R} x_r \otimes y_r \otimes z_r$$

The optimal rank-$R$ decomposition of $X$ is determined by minimizing the error between $X$ and the reconstructed tensor $\widehat{X}$. We use the elbow function heuristic to select the best-fit rank from a range of 2–12 factors. This approach seeks to balance the complexity of the model against the fidelity of reconstruction. Missing values in $X$ are ignored when calculating the loss.

### RNAflux: unsupervised spatial embedding and subcellular domain quantization

To generate RNAflux embeddings, first, a set of query coordinates are generated tiling across the cell area on a uniform grid. This effectively downsamples the original data units (pixels) resulting in much fewer samples to compute embeddings. Let $Q = \{q_1, q_2, ..., q_n\}$ be the set of query coordinates and $q_i$ denote the $i^{th}$ query coordinate. For the MERFISH U2-OS dataset, a step size of 10 data units (pixels) was used to generate the uniform grid. For the iPSC-derived cardiomyocytes, a step size of 5 data units

was used. Each query point is assigned an expression vector, counting the abundance of each gene within a fixed radius of 40 and 50 data units respectively. Each expression vector is normalized to sum to one, converting the expression vector to a composition vector, denoted as $v_i$. Similarly, the cell composition vector $x_j$ for the $j^{th}$ cell is calculated by normalizing the total cell expression to sum to one. The RNAflux embedding $r_i$ at query coordinate $q_i$ is defined as the difference between the $v_i$ and $x_j$ divided by $\sigma_v$, the vector of standard deviation of composition vectors $v$.

$$r_i = \frac{v_i - x_j}{\sigma_v}$$

The RNAflux embedding serves as an interpretable spatial gene embedding that quantifies highly local fluctuations in gene composition. Dimensional reduction of the embeddings is performed using truncated singular value decomposition (SVD). Truncated SVD was chosen over PCA to better handle large but sparse data. Embeddings were reduced to the top 10 components. To assign domains, self-organizing maps (SOM) as implemented in MiniSom [82] were used for low-rank quantization of query embeddings. In an analysis of the MERFISH dataset, SOMs of size $1 \times k$ were fit across a range of 2 to 12; the best model was determined using the elbow method heuristic to evaluate quantization error. Similarly, domains were determined for the cardiomyocyte spatial transcriptomics data by fitting the vehicle and treatment samples separately, for $k$ across a range of 2 to 8. The elbow method heuristic determined an optimal $k$ of 6; subsequently, a $k$ of 4 was used for further analysis for ease of interpretation.

### RNAflux: visualizing spatial embeddings

The top 3 principal components of the RNAflux embeddings are transformed to map red, green, and blue values respectively. Embeddings are first quantile normalized and scaled to a minimum of 0.1 and 0.9 to avoid mapping extreme quantiles to white and black. These values are then used for red, green, and blue color channels. To map the downsampled grid back to the original data units, linear interpolation was used to rescale the computed color values and fill the space between the uniform grid points.

### RNAflux: enrichment of locale-specific transcriptomes derived by APEX-seq

The enrichment score for each pixel is calculated by first taking the weighted sum of its RNAflux embedding and locale-specific log fold-change values as implemented by the decoupler tool [83]. Scores for pixels within a given cell are normalized against a null distribution constructed via random permutations of the input embeddings, to produce *z*-scaled enrichment scores. Fluxmap domain enrichment scores are simply obtained by taking the mean score of all pixels within the boundary of each domain. Fluxmap domain overlaps are computed by counting the fraction of pixels within the boundary of each domain with a positive enrichment score.

### MERFISH of U2-OS cells

*MERFISH sample preparation.* MERFISH measurements of 130 genes with five non-targeting blank controls were done as previously described, using the published encoding [44] and readout probes [84]. Briefly, U2-OS cells were cultured on 40 mm #1.5

coverslips that are silanized and poly-L-lysine coated [44] and subsequently fixed in 4% (vol/vol) paraformaldehyde in $1 \times$ PBS for 15 min at room temperature. Cells were then permeabilized in 0.5% Triton X-100 for 10 min at room temperature and washed in $1 \times$ PBS containing Murine RNase Inhibitor (NEB M0314S). Cells were preincubated with hybridization wash buffer (30% (vol/vol) formamide in $2 \times$ SSC) for 10 min at room temperature with gentle shaking. After preincubation, the coverslip was moved to a fresh 60-mm petri dish and residual hybridization wash buffer was removed with a Kimwipe lab tissue. In the new dish, 50 uL of encoding probe hybridization buffer ($2 \times$ SSC, 30% (vol/vol) formamide, 10% (wt/vol) dextran sulfate, 1 mg ml$^{-1}$ yeast tRNA, and a total concentration of 5 uM encoding probes and 1 µM of anchor probe: a 15-nt sequence of alternating dT and thymidine-locked nucleic acid (dT +) with a $5'$-acrydite modification (Integrated DNA Technologies). The sample was placed in a humidified 37 °C oven for 36 to 48 h, then washed with 30% (vol/vol) formamide in $2 \times$ SSC for 20 min at 37 °C, 20 min at room temperature. Samples were post-fixed with 4% (vol/vol) paraformaldehyde in $2 \times$ SSC and washed with $2 \times$ SSC with murine RNase inhibitor for 5 min. The samples were finally stained with an Alexa 488-conjugated anchor probe-readout oligo (Integrated DNA Technologies) and DAPI solution at 1 µg/ml.

### MERFISH imaging

MERFISH measurements were conducted on a home-built system as described in Huang et al. [84].

### MERFISH spot detection

Individual RNA molecules were decoded in MERFISH images using MERlin v0.1.6 [85]. Images were aligned across hybridization rounds by maximizing phase cross-correlation on the fiducial bead channel to adjust for drift in the position of the stage from round to round. Background was reduced by applying a high-pass filter and decoding was then performed per-pixel. For each pixel, a vector was constructed of the 16 brightness values from each of the 16 rounds of imaging. These vectors were then L2 normalized and their Euclidean distances to each of the L2 normalized barcodes from the MERFISH codebook were calculated. Pixels were assigned to the gene whose barcode they were closest to, unless the closest distance was greater than 0.512, in which case the pixel was not assigned a gene. Adjacent pixels assigned to the same gene were combined into a single RNA molecule. Molecules were filtered to remove potential false positives by comparing the mean brightness, pixel size, and distance to the closest barcode of molecules assigned to blank barcodes to those assigned to genes to achieve an estimated misidentification rate of 5%. The exact position of each molecule was calculated as the median position of all pixels consisting of the molecule.

### MERFISH image segmentation

Cellpose v1.0.2 [70] was used to perform image segmentation to determine the boundaries of cells and nuclei. The nuclei boundaries were determined by running Cellpose with the "nuclei" model using default parameters on the DAPI stain channel of the pre-hybridization images. Cytoplasm boundaries were segmented with the "cyto" model and default parameters using the polyT stain channel. RNA molecules identified by MERlin

were assigned to cells and nuclei by applying these segmentation masks to the positions of the molecules.

### iPSC cardiac differentiation and doxorubicin treatment

Matrigel (Corning, cat # 354,277)-coated plates were used to culture iPSCs with mTESR Plus human iPSC medium (StemCell Technologies, cat # 100–0276) in a humidified incubator at 37 °C with 5% $CO_2$. iPSCs were dissociated with Gentle Cell Dissociation Reagent (StemCell Technologies, cat # 100–0485) and passaged with mTESR Plus medium and 10uM ROCK inhibitor (Tocris, cat #1254) at a ratio of 1:12. mTESR plus medium was replaced every other day until the cells reached 80% confluency for maintenance and replating, or 90% confluency for cardiac differentiation utilizing a chemically defined protocol [86]. On day 0 of cardiac differentiation, cells were treated with 6 μM CHIR99021 (Selleck Chem, cat # S1263) in RPMI 1640 media (Gibco, cat # 11,875) and B27 minus insulin supplement (Thermo Fisher, cat # A1895601). On day 2, CHIR was removed, and cells were cultured with RPMI 1640 media and B27 minus insulin supplement (Thermo Fisher, cat # A18956). On day 3, media was replaced with RPMI media containing B27 minus insulin supplement and 5 μM Wnt-C59 (Cellagen Technologies, cat # C7641-2 s). On days 5, 7, and 9, media was replaced with RPMI media containing B27 insulin supplement (Thermo Fisher, cat # 17,504). On days 11 and 13, media was replaced with RPMI 1640 media without glucose (Thermo Fisher, cat # 11,879,020) containing B27 insulin supplement for purification of cardiomyocytes. From days 15 onward, the cells were cultured in RPMI 1640 media containing B27 supplement which was changed every other day until the cells reached day 30 for replating. For replating, cells were incubated in $10 \times$ TrypLE (Thermo Fisher, cat # A1217701) for 12 min at 37 °C, neutralized with equal volumes of RPMI 1640 media containing B27 supplement with 20% FBS (Gibco, cat # 26,140–079), gently dissociated by pipetting, then spun down and resuspended for replating in RPMI 1640 media containing B27 supplement with 20% FBS. The next day, the cell media was replaced with RPMI 1640 media containing B27 supplement which was replaced with fresh media every other day. On day 48, the cells were replated onto chamber slides (Ibidi, cat # 80,826) as described above and recovered for 10 days before doxorubicin treatments began (MedChemExpress, cat # HY-15142). On day 60, doxorubicin treatments concluded, and the cells underwent methanol fixation.

### Molecular *cartography*

#### *Cultured cell processing*

After doxorubicin treatment, cardiomyocytes were washed with PBS ($1 \times$) twice and fixed in methanol ($-20$ °C) for 10 min. After fixation, Methanol was aspirated and cells were dried and stored at $-80$ °C until use. The samples were used for Molecular Cartography™ (100-plex combinatorial single-molecule fluorescence in-situ hybridization) according to the manufacturer's instructions *Day 1: Molecular Preparation Protocol* for cells, starting with the addition of buffer DST1 followed by cell priming and hybridization. Briefly, cells were primed for 30 min at 37 °C followed by overnight hybridization of all probes specific for the target genes (see below for probe design details and target list). Samples were washed the next day to remove excess probes and fluorescently tagged

in a two-step color development process. Regions of interest were imaged as described below and fluorescent signals removed during decolorization. Color development, imaging, and decolorization were repeated for multiple cycles to build a unique combinatorial code for every target gene that was derived from raw images as described below.

### Probe design

The probes for 100 genes were designed using Resolve's proprietary design algorithm. Briefly, the probe design was performed at the gene level. For every targeted gene, all full-length protein-coding transcript sequences from the ENSEMBL database were used as design targets if the isoform had the GENCODE annotation tag "basic" [87, 88]. To speed up the process, the calculation of computationally expensive parts, especially the off-target searches, the selection of probe sequences was not performed randomly, but limited to sequences with high success rates. To filter highly repetitive regions, the abundance of k-mers was obtained from the background transcriptome using Jellyfish [89]. Every target sequence was scanned once for all k-mers, and those regions with rare k-mers were preferred as seeds for full probe design. A probe candidate was generated by extending a seed sequence until a certain target stability was reached. A set of simple rules was applied to discard sequences that were found experimentally to cause problems. After these fast screens, the remaining probe candidates were mapped to the background transcriptome using ThermonucleotideBLAST [90] and probes with stable off-target hits were discarded. Specific probes were then scored based on the number of on-target matches (isoforms), which were weighted by their associated APPRIS level [91], favoring principal isoforms over others. A bonus was added if the binding site was inside the protein-coding region. From the pool of accepted probes, the final set was composed by picking the highest-scoring probes. Probes with catalog numbers can be found in Additional File 3: Table S3.

### Imaging

Samples were imaged on a Zeiss Celldiscoverer 7, using the $50 \times$ Plan Apochromat water immersion objective with an NA of 1.2 and the $0.5 \times$ magnification changer, resulting in a $25 \times$ final magnification. Standard CD7 LED excitation light source, filters, and dichroic mirrors were used together with customized emission filters optimized for detecting specific signals. Excitation time per image was 1000 ms for each channel (DAPI was 20 ms). A z-stack was taken at each region with a distance per z-slice according to the Nyquist-Shannon sampling theorem. The custom CD7 CMOS camera (Zeiss Axiocam Mono 712, 3.45 µm pixel size) was used. For each region, a z-stack per fluorescent color (two colors) was imaged per imaging round. A total of 8 imaging rounds were done for each position, resulting in 16 z-stacks per region. The completely automated imaging process per round was realized by a custom Python script using the scripting API of the Zeiss ZEN software (Open application development).

### Image processing and spot segmentation

As a first step, all images were corrected for background fluorescence. A target value for the allowed number of maxima was determined based on the area of the slice in µm$^2$ multiplied by the factor 0.5. This factor was empirically optimized. The brightest maxima per plane were determined, based upon an empirically optimized threshold. The number and

location of the respective maxima were stored. This procedure was done for every image slice independently. Maxima that did not have a neighboring maximum in an adjacent slice (called z-group) were excluded. The resulting maxima list was further filtered in an iterative loop by adjusting the allowed thresholds for (Babs-Bback) and (Bperi-Bback) to reach a feature target value (Babs: absolute brightness, Bback: local background, Bperi: background of periphery within 1 pixel). These feature target values were based on the volume of the 3D image. Only maxima still in a group of at least 2 after filtering were passing the filter step. Each z-group was counted as one hit. The members of the z-groups with the highest absolute brightness were used as features and written to a file. They resemble a 3D-point cloud. To align the raw data images from different imaging rounds, images had to be registered. To do so, the extracted feature point clouds were used to find the transformation matrices. For this purpose, an iterative closest point cloud algorithm was used to minimize the error between two point clouds. The point clouds of each round were aligned to the point cloud of round one (reference point cloud). The corresponding point clouds were stored for downstream processes. Based upon the transformation matrices, the corresponding images were processed by a rigid transformation using trilinear interpolation. The aligned images were used to create a profile for each pixel consisting of 16 values (16 images from two color channels in 8 imaging rounds). The pixel profiles were filtered for variance from zero normalized by the total brightness of all pixels in the profile. Matched pixel profiles with the highest score were assigned as an ID to the pixel. Pixels with neighbors having the same ID were grouped. The pixel groups were filtered by group size, number of direct adjacent pixels in group, and number of dimensions with a size of two pixels. The local 3D-maxima of the groups were determined as potential final transcript locations. Maxima were filtered by the number of maxima in the raw data images where a maximum was expected. The remaining maxima were further evaluated by the fit to the corresponding code. The remaining maxima were written to the results file and considered to resemble transcripts of the corresponding gene. The ratio of signals matching to codes used in the experiment and signals matching to codes not used in the experiment were used as estimation for specificity (false positives). The algorithms for spot segmentation were written in Java and are based on the ImageJ library functionalities. Only the iterative closest point algorithm is written in C++ based on the libpointmatcher library (https://github.com/ethz-asl/libpointmatcher).

### *Image segmentation*

Cellpose v1.0.2 [70] was used to perform image segmentation to determine the boundaries of nuclei. The nuclei boundaries were determined by running Cellpose with the "nuclei" model using default parameters on the DAPI stain channel of the pre-hybridization images. Cytoplasm boundaries were determined with ClusterMap [24] using spot coordinates.

### Review history

The review history is available as Additional file 5.

### Peer review information

## Supplementary Information

---

**Additional file 1:** Supplementary Figures and Table S1. Supplementary figures and machine learning classifier feature descriptions.

**Additional file 2: Table S2.** Hyperoptimization parameters and classifier architectures.

**Additional file 3: Table S3.** Cardiomyocyte gene panel for Molecular Cartography experiment.

**Additional file 4: Table S4.** Manual annotation data for RNA localization patterns

**Additional file 5.** Review history

---

### Acknowledgements

We thank members of the Yeo lab, Carter lab, Michelle Franc Ragsac, Erick Armingol, and Nate Lewis for helpful discussions and feedback on the manuscript.

### Authors' contributions

C.K.M, N.A., and G.W.Y. conceptualized the project. C.K.M. and N.A. co-developed the software. C.K.M. and D.L. trained the classification model for subcellular localization. C.K.M., N.A., and D.L. manually annotated data for benchmarking model performance. C.K.M., N.A., and G.P. performed data preprocessing and analysis. A.M., C.K., Y.H., and Q.Z. generated the MERFISH experiment. N.L. designed the gene panel and cultured the cardiomyocytes. A.C. and E.L. aided multimodal spatial analyses. C.K.M., N.A., H.C., and G.W.Y. wrote the manuscript. H.C. and G.W.Y. supervised the project.

### Funding

C.K.M. is supported by the National Science Foundation Graduate Research Fellowship under Grant No. (DGE-2038238). N.A. was partially supported by NIH Training Grant T32 GM008666. This work was partially supported by National Institutes of Health grants NS103172, MH107367, AI132122, AI123202, AG069098, HG004659, and HG009889 to G.W.Y. G.W.Y. is also supported by an Allen Distinguished Investigator Award, a Paul G. Allen Frontiers Group advised grant of the Paul G. Allen Family Foundation. A.J.C. and E.L. acknowledge support from the Chan Zuckerberg Initiative (CZF2019-002448) and the Knut and Alice Wallenberg Foundation (KAW 2021.0346) to E.L.

### Availability of data and materials

The datasets (seqFISH+, MERFISH, and Molecular Cartography datasets) supporting the conclusions of this article are available in the Figshare repository, https://doi.org/10.6084/m9.figshare.c.6564043.v1 [92], and are also accessible through the Bento Python package.
The source code for Bento is available on the GitHub repository: https://github.com/ckmah/bento-tools [93]. Analysis code for generating figures can be found at https://github.com/ckmah/bento-manuscript [94] and is archived at https://doi.org/10.5281/zenodo.10815484 [95]. Documentation for Bento can be found here: http://bento-tools.readthedocs.io.

## Declarations

### Ethics approval and consent to participate
Not applicable.

### Consent for publication
Not applicable.

### Competing interests

G.W.Y. is a co-founder, member of the board of directors, equity holder, and paid consultant for Locanabio (until 12/31/2023) and Eclipse Bioinnovations, and a Scientific Adviser and paid consultant to Jumpcode Genomics. G.W.Y. is a Distinguished Visiting Professor at the National University of Singapore. The terms of these arrangements have been reviewed and approved by the University of California, San Diego, in accordance with its conflict-of-interest policies. The authors declare no other competing interests.

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

## 