## [**Additional file 5.** Review history · Genome Biology]

Review History

First round of review

Reviewer 1

Were you able to assess all statistics in the manuscript, including the appropriateness of statistical tests used? Yes, and I have assessed the statistics in my report.

Were you able to directly test the methods? No.

Comments to author:

The manuscript titled "Bento: A toolkit for subcellular analysis of spatial transcriptomics data" by K. Mah et al. introduces Bento, a subcellular analysis toolkit featuring RNA localization annotation (RNAforest), subcellular domain clusters (RNAcoloc), and gene-gene colocalization (RNAflux), drawing from concepts in current cell-cell communication analysis. The authors offer fresh insights into single-molecular data using segmentation and FISH tools, setting the stage for subcellular spatial transcriptome data analysis. This entails numerous models and quantitative metrics. However, the definition and application of these models need to be organized and elucidated for reader comprehension, given the dynamic nature of RNA molecule localization within a cell in most cases. The robustness and biological significance of this level of localization should be explained with more substantial evidence. Furthermore, the importance and value of RNA localization and colocalization have not been fully scrutinized or validated. Additionally, single-molecular data holds great potential for shedding light on recycling molecules and signaling cascades, particularly in molecular trafficking and interaction between different organelles. The potential contribution of Bento to understanding this process warrants discussion.

Minor concerns:

1. What are the 13 spatial features? Figure 2a only lists 4 categories, with no specific quantitative indicators.
2. Many metrics lack calculation instructions and equations, such as the aspect ratio, which denotes the ratio of the major axis to the minor axis. How should the irregular shape of cells be defined?
3. In Figure S1e, why is the AUROC value of the test data higher than the value of the validation data? Are there different distributions in the data sets?
4. In Figure 2e, what do the other 2 grey bars represent, in addition to the denoted 5 labels?
5. In Figure 3d, the specificity of these 4 factors is not clearly defined. How should these 4 factors be understood and defined?
6. Regarding RNAflux embedding, what is the corresponding cell composition? For instance, what is meant by "the difference between the query composition and its corresponding cell composition...?"

7. For some imaging-based methods, e.g., Xenium, the RNA molecules were in situ amplified, creating signal spots that may become overcrowded. How can Bento segment the transcripts from the overlapping signals?
8. In the current subcellular imaging-based spatial transcriptome, most of the target panel genes are related to classifying different cell types. The authors should be cautious that the model may be biased by the specific gene-set used in the dataset.
9. The figure legends should specify the n numbers and statistical tests where appropriate.

Reviewer 2

Were you able to assess all statistics in the manuscript, including the appropriateness of statistical tests used? There are no statistics in the manuscript.

Were you able to directly test the methods? No.

Comments to author:

The paper presents a new tool, Bento for analyzing spatial transcriptomics data at the subcellular level. Bento, a Python toolkit, addresses a significant gap in spatial transcriptomics by enabling analysis at the subcellular scale. This is a notable advancement over existing methods that focus on multicellular scale analysis. The toolkit can ingest data from various spatial transcriptomics technologies and is compatible with single-molecule resolution data. It integrates well with the Severse ecosystem, allowing it to use other omics analysis tools. Bento introduces three essential analytical methods: RNAforest for annotating RNA localization patterns, RNacoloc for analyzing RNA colocalization, and RNAflux for identifying distinct subcellular domains. This paper is timely, given the new technology. The tool is open source. The paper is mostly well written. Overall, it is an excellent paper.

===

Please consider the following questions:

- Are the methods appropriate to the aims of the study, are they well described, and are necessary controls included? If not, please specify what is required.

Yes.

- Are the conclusions adequately supported by the data shown? If not, please explain

Mostly. The training data used are 10,000 simulated samples, which may be significantly different from real data. The generalizability of the method is unclear as no related computational results are provided. Some discussion of this limitation should be given.

- Are sufficient details provided to allow replication and comparison with related analyses that may have been performed? If not, please specify what is required.

The computational methods are not described in sufficient details for replication. Some more details should be given. For example, machine learning methods used should provide features, architectures, etc.

- Does the work represent a significant advance over previously published studies?

Yes.

- Is the paper of broad interest to others in the field, or of outstanding interest to a broad audience of biologists?

Yes, it is of interest to a broad audience of biologists.

Reviewer 3

Were you able to assess all statistics in the manuscript, including the appropriateness of statistical tests used? Yes, and I have assessed the statistics in my report.

Were you able to directly test the methods? Yes.

Comments to author:

The manuscript entitled "Bento: A Toolkit for Subcellular Analysis of Spatial Transcriptomics Data" presents Bento, a Python toolkit designed for spatial analysis at the subcellular scale. It focuses on the spatial organization of molecules within cells and aims to expand the capabilities of current computational methods that mostly deal with cell-level analysis. Bento processes molecular coordinates and segmentation boundaries, allowing for defining subcellular domains, annotating localization patterns, and quantifying gene-gene colocalization. The toolkit's efficacy is demonstrated through applications to various datasets, including cardiomyocytes treated with doxorubicin, highlighting its potential in subcellular resolution spatially resolved single-cell omics analysis. Overall, Bento is a very well-written and structured paper. The analysis, although lacking brand-new method development, is very well conducted, and the figures are well designed and delivered. Except for some comments I will detail in the following paragraphs, this work is an excellent contribution to the spatial transcriptomics field.

Major comments:

1. The author highlighted that their approach is generally applicable to all kinds of subcellular resolution spatial transcriptomics. However, the main analysis of this work is done on the MERFISH dataset of U2-OS cells and the Molecular Cartography dataset of cardiomyocytes. It would be especially interesting to check how this approach performs on a few other data types, for example, the STARmap dataset or other high-definition sequencing-based datasets, including seqScope, PIXEL-seq, and Stereo-seq datasets. In particular, the author brought up a nice point that mislocalization of RNA may be related to different diseases. It thus may be interesting to check the StarMap dataset of Alzheimer's disease: "Integrative in situ mapping of single-cell transcriptional states and tissue histopathology in a mouse model of Alzheimer's disease" | Nature Neuroscience.

2. One reason previous spatial transcriptomics analysis approaches don't look into the subcellular information is probably the result of the difficulty in handling the subcellular information of each molecule within each cell. Bento uses anndata to deposit the data but doesn't support subcellular information for each molecule within each cell. I would love to see how the author deals with this and, importantly, whether they can come up with a nice solution to encode both the subcellular information for each molecule and spatial information of each cell.

3. Another reason previous spatial transcriptomics analysis approaches don't look into the subcellular information is probably the fact that the measurement of subcellular information requires high spatial resolution and precision. At this moment, it is unclear what the spatial resolution is for each spatial transcriptomics technology. It will be important for the author to assess the level of spatial precision for each major spatial transcriptomics technology. Importantly, they should quantify the level of RNA diffusion during the imaging or sequencing library preparation step. Furthermore, it will be important to discuss the level of subcellular analysis that can be conducted for each technology, given their subcellular spatial resolution.

4. It is a nice idea to compare the data from APEX-seq with the MERFISH data. However, APEX-seq is based on bulk measurements. It will be very interesting to see whether the subcellular RNA distribution differs across different cell types, etc.

5. Tensor decomposition for compartment-specific colocalization: I don't follow the mathematical description in this section. Can you explain more about what the dataset tensor X is? And what is the rank-one 3-way tensors?

6. Make sure to use either capitalized or lowercase a/b/c in figures and figure captions consistently. Currently, all figures use lowercase, but captions use capital letters.

7. Figure 1:

- What does the color in panel D mean? You may want to explicitly mention that the same set of colors from panel A is consistently used throughout the figure.

- It is weird to see that although ANKRD52, CEP250, etc., are treated as "None" genes, they show enrichment in the cytoplasmic group. Also, please label the legend for the colormaps in panel H/I.

8. Clustermap is known to have low performance for single-cell segmentation. Can you please try Baysoy or Spateo for single-cell segmentation?

9. The data shown in the last figure is interesting. However, the analysis for this dataset is rather shallow. For example, we can ask the following questions:

- Can you find some genes that don't show expression changes before and after drug treatment but have a high degree of shifts in terms of subcellular distribution?

- In panels F/G, you highlighted the depletion of RNA within the cytoplasm. Can we find genes that show the opposite pattern? What genes show high depletion in the nucleus after the Dox treatment?

Minor comments:

1. Some language may need improvement. For example, I don't follow (in the section RNAflux: xxxx, last paragraph) why the computation of a gene composition vector will generate a spatial computation gradient across entire cells. In your figure 4a, the RNAflux embedding shows different colors, and the colors are generated from the top three principal components. Is it precise to call the color difference the computation gradient? I would suggest providing more details in that sentence.
2. The font size for the CLQ score and spatial domains in Figure 4b is too small. A minimal font size of 6 is needed.
3. Font sizes for Figure 5e are too small as well.

Revisions

Responses to Reviewer #1

The manuscript titled "Bento: A toolkit for subcellular analysis of spatial transcriptomics data" by K. Mah et al. introduces Bento, a subcellular analysis toolkit featuring RNA localization annotation (RNAforest), subcellular domain clusters (RNAcoloc), and gene-gene colocalization (RNAflux), drawing from concepts in current cell-cell communication analysis. The authors offer fresh insights into single-molecular data using segmentation and FISH tools, setting the stage for subcellular spatial transcriptome data analysis. This entails numerous models and quantitative metrics. However, the definition and application of these models need to be organized and elucidated for reader comprehension, given the dynamic nature of RNA molecule localization within a cell in most cases. The robustness and biological significance of this level of localization should be explained with more substantial evidence. Furthermore, the importance and value of RNA localization and colocalization have not been fully scrutinized or validated. Additionally, single-molecular data holds great potential for shedding light on recycling molecules and signaling cascades, particularly in molecular trafficking and interaction between different organelles. The potential contribution of Bento to understanding this process warrants discussion.

Minor Concerns

- *What are the 13 spatial features? Figure 2a only lists 4 categories, with no specific quantitative indicators.*

The spatial features are described in detail in Supplementary Table 1. In the main text, the table is referenced in the sentence immediately after Figure 2a is referenced. An additional reference was added to the figure legend for Figure 2a.

- *In Figure S1e, why is the AUROC value of the test data higher than the value of the validation data? Are there different distributions in the data sets?*

We have renamed supplementary figure titles to "AUROC Simulated Test" and "AUROC Real-World Validation" to clarify which datasets "test" and "validation" refer to. We also added an additional statement to the main text to clarify the "validation data":

Validation performance on manual annotation of subsets of both datasets show that RNAforest generalizes well despite biological and technical differences (**Methods, Supp. Fig. 1b**).

The "test data" is defined in the second paragraph of the "RNAforest" section of the main text:

In total, we simulated 2,000 samples per class for a total of 10,000 samples (**Methods**). We used 80% of the simulated data for training and held out the remaining 20% for testing.

- *In Figure 2e, what do the other 2 grey bars represent, in addition to the denoted 5 labels?*

The two right-most grey bars represent transcripts annotated with two labels. As indicated by the dots underneath, the first bar denotes "cell edge" and "none" labeled transcripts and the second bar denotes "cytoplasmic" and "none" labeled transcripts. The Figure 2 legend has been updated to the following:

D) and **E)** are UpSet plots showing the proportion of measured transcripts assigned to each label.

- *In Figure 3d, the specificity of these 4 factors is not clearly defined. How should these 4 factors be understood and defined?*

The paragraph interpreting RNAcoloc results has been rewritten to define the 4 factors more succinctly. Additionally, Figures 3C and 3D has been revised to accommodate the updated text. Below is the updated 5th paragraph under "RNAcoloc: An approach for context-specific RNA colocalization":

Applied to the U2-OS dataset, RNAcoloc decomposes RNA colocalization into 4 factors. Examining factor loadings indicate two distinct subpopulations of cells with compartment-specific colocalization behaviors; cluster 1 cells exhibit uniform (Factor 0) and cytoplasmic (Factor 3) colocalization, while cluster 2 cells show nuclear (Factor 1) and cytoplasmic colocalization (Factor 2) (Fig. 3C-3D). Factor 3 describes colocalization of gene pairs in the cytoplasm of cluster 1 cells, especially a number of genes that attract PIK3CA transcripts. While little is known about PIK3CA RNA interactions, the PI3K pathway regulates mitotic organization, including the regulation of dynein and dynactin motor proteins. DYNC1H1 is among the top genes attracting PIK3CA, and specifically encodes cytoplasmic dynein, a motor protein critical for spindle formation and chromosomal segregation in mitosis[54]. This hints that not only is compartmental localization of RNA linked to the cell cycle[45], but RNA-RNA interactions may play a role as well. In cluster 2 cells, MALAT1 attracts CNR2 transcripts more than expected in the cytoplasm. Even though MALAT1 is canonically abundantly localized to the nucleus, this demonstrates that the CLQ score identifies gene pairs colocalizing more than expected despite disproportionate expression of MALAT1 relative to CNR2, whereas other approaches seem confounded by large differences in expression[44].

- *Regarding RNAflux embedding, what is the corresponding cell composition? For instance, what is meant by "the difference between the query composition and its corresponding cell composition..."?*

We have added notation to better illustrate calculations for the RNAflux embedding. See our updated methods section for RNAflux. In regards to this particular question, the cell composition is the normalized total expression of each gene across the entire single cell, as opposed to the query composition which is the normalized expression of each gene at a specific region of a cell.

- *For some imaging-based methods, e.g., Xenium, the RNA molecules were in situ amplified, creating signal spots that may become overcrowded. How can Bento segment the transcripts from the overlapping signals?*

Bento does not perform spot detection and decoding and leaves it up to the specific technology platform. Just as each platform has unique chemistry, platforms have their own specific pipelines for handling image analysis. All commercial platforms e.g. 10X Xenium, Nanostring CosMx, Vizgen MERSCOPE have proprietary pipelines and do not share raw imaging data. Spot detection and decoding is not in the scope of Bento's functionality. Bento is downstream of these processes, enabling subcellular localization analysis of processed data.

- *In the current subcellular imaging-based spatial transcriptome, most of the target panel genes are related to classifying different cell types. The authors should be cautious that the model may be biased by the specific gene-set used in the dataset.*

We agree with the reviewers and thank them for emphasizing this limitation. We have added the text below regarding the influence of panel design on RNAflux.

As most commercial target panels are largely composed of marker genes for cell type identification, RNAflux embeddings should be interpreted carefully, especially if transcripts show little spatial variation in subcellular localization.

- *The figure legends should specify the n numbers and statistical tests where appropriate.*

Figure 2i: A legend for stars denoting p-values was added. The statistical test and FDR corrections used are stated in the figure legend.

Figure 3b: Statistical test, FDR correction used and number of samples per category are stated in the figure legend.

Figure 5b: Statistical test, FDR correction used and number of samples per condition are stated in the figure legend.

Figure 5e: Added distance metric used for measuring shifts in gene localization.

Responses to Reviewer #2

The paper presents a new tool, Bento for analyzing spatial transcriptomics data at the subcellular level. Bento, a Python toolkit, addresses a significant gap in spatial transcriptomics by enabling analysis at the subcellular scale. This is a notable advancement over existing methods that focus on multicellular scale analysis. The toolkit can ingest data from various spatial transcriptomics technologies and is compatible with single-molecule resolution data. It integrates well with the Scverse ecosystem, allowing it to use other omics analysis tools. Bento introduces three essential analytical methods: RNAforest for annotating RNA localization patterns, RNacoloc for analyzing RNA colocalization, and RNAflux for identifying distinct subcellular domains. This paper is timely, given the new technology. The tool is open source. The paper is mostly well written. Overall, it is an excellent paper.

Please consider the following questions:

1. Are the methods appropriate to the aims of the study, are they well described, and are necessary controls included? If not, please specify what is required.

Yes.

1. Are the conclusions adequately supported by the data shown? If not, please explain.
 - *Mostly. The training data used are 10,000 simulated samples, which may be significantly different from real data. The generalizability of the method is unclear as no related computational results are provided. Some discussion of this limitation should be given.*

We thank the reviewer for seeking clarification on the validation of RNAforest. Indeed, generalizability of RNAforest can be a concern particularly when applied to tissue data with poor segmentation quality as opposed to cell culture. However, true evaluation of model generalizability can only be done with annotated ground truth data. In fact, we already made efforts to this end already as can be seen in Supplementary Table 4 and in the Methods section “RNAforest: Manual annotation of true biological validation data”. An excerpt of that section:

Using 3 individual annotators, we annotated the same 600 samples across both datasets, keeping samples with 2 or more annotator agreements as true annotations, resulting in 165 annotated seqFISH+ samples and 238 annotated MERFISH samples (403 total). We used Cohen’s kappa coefficient[76] to calculate agreement between pairs of annotators for each label yielding an overall coefficient of 0.602. We found that pairwise agreement between annotators across labels was fairly consistent ranging between 0.588 and 0.628, while label-specific agreement varied more, ranging between 0.45 and 0.72 (Supp. Table 4).

We have also linked the previously published scripts for RNAforest training data generation, and describe in detail the parameters used when running the scripts (Methods). To some degree, the concern for the generalizability of RNAforest comes from the observation that different cell types in different cultures or tissues have very different morphologies and shapes. It is critical, then, to note that RNAforest does not predict subcellular localization patterns explicitly from the morphology of the cell. Rather, 13 different spatial features (see Supp Table 1) are computed for the purpose of abstracting the spatial characteristics of RNA localization in relation to the cell and nuclear boundaries away from the actual shape of the cell. For example, regardless of the resolution (number of vertices) of a cell segmentation mask, the distance of the center of mass of an RNA point cloud from the center of mass of a cell will not vary. We undertook this strategy to maximize generalizability, particularly because we know that cell segmentation is not a solved problem.

1. Are sufficient details provided to allow replication and comparison with related analyses that may have been performed? If not, please specify what is required.
 - *The computational methods are not described in sufficient details for replication. Some more details should be given. For example, machine learning methods used should provide features, architectures, etc.*

We provide detailed feature inputs as well as model architectures and hyperparameters for RNAforest in Supplementary Tables 1 & 2 as well as in our methods. We have added a reference to Supplementary Table 2 and Methods in the main text to make it easier to find. For RNAColoc and RNAflux, we have substantially updated the methods section to include more detailed mathematical notation and steps of each algorithm. We have provided all scripts, data and methods needed to reproduce figures. The repository is linked under the “Code Availability” and “Data Availability” section as required by the journal.

1. Does the work represent a significant advance over previously published studies?
 - Yes.
1. Is the paper of broad interest to others in the field, or of outstanding interest to a broad audience of biologists?
 - *Yes, it is of interest to a broad audience of biologists.*

Responses to Reviewer #3

The manuscript entitled "Bento: A Toolkit for Subcellular Analysis of Spatial Transcriptomics Data" presents Bento, a Python toolkit designed for spatial analysis at the subcellular scale. It focuses on the spatial organization of molecules within cells and aims to expand the capabilities of current computational methods that mostly deal with cell-level analysis. Bento processes molecular coordinates and segmentation boundaries, allowing for defining subcellular domains, annotating localization patterns, and quantifying gene-gene colocalization. The toolkit's efficacy is demonstrated through applications to various datasets, including cardiomyocytes treated with doxorubicin, highlighting its potential in subcellular resolution spatially resolved single-cell omics analysis. Overall, Bento is a very well-written and structured paper. The analysis, although lacking brand-new method development, is very well conducted, and the figures are well designed and delivered. Except for some comments I will detail in the following paragraphs, this work is an excellent contribution to the spatial transcriptomics field.

Major Comments

- *The author highlighted that their approach is generally applicable to all kinds of subcellular resolution spatial transcriptomics. However, the main analysis of this work is done on the MERFISH dataset of U2-OS cells and the Molecular Cartography dataset of cardiomyocytes. It would be especially interesting to check how this approach performs on a few other data types, for example, the STARmap dataset or other high-definition sequencing-based datasets, including seqScope, PIXEL-seq, and Stereo-seq datasets. In particular, the author brought up a nice point that mislocalization of RNA may be related to different diseases. It thus may be interesting to check the StarMap dataset of Alzheimer's disease: "Integrative in situ mapping of single-cell transcriptional states and tissue histopathology in a mouse model of Alzheimer's disease" | Nature Neuroscience.*

We thank the reviewer for seeking further application of our approach to other dataset types. As the reviewer points out, in the manuscript we have already provided analyses on data generated by MERFISH, seqFISH+, and Molecular Cartography. We have provided easy to use API endpoints for users to ingest data generated by any platform to leverage the entire suite of tools in Bento. While an exhaustive demonstration of Bento on every platform is well outside the scope of this paper, it is true that our previous version of the manuscript only analyzed data from platforms utilizing multiplexed smFISH imaging on homogenous cell types. Thus, we have now added an analysis using RNAflux on a previously published breast cancer tissue dataset generated on the 10x Xenium platform--which utilizes padlock probe-based targeting as opposed to smFISH. The added analysis can be found in Supplementary Figure 4 along with a paragraph describing the results in the results section titled "RNAflux: Unsupervised semantic segmentation of subcellular domains in single cells":

The most common application for spatial transcriptomics is in mapping heterogeneous cell types in large tissue samples. This presents several challenges. First, the panels for these experiments are weighted heavily towards cell type markers determined by single-nuclei RNA-seq, making the intracellular variability in expression not a guarantee. Second, substantial intracellular heterogeneity that can skew the low-rank embedding. To explore the applicability of RNAflux on tissue, we applied it to a previously published breast cancer tissue dataset generated by 10x Xenium.[58] We successfully reproduced the identification of unique cell types (Fig. S4A & S4B). By applying RNAflux, we find that fluxmaps 1-3 show different enrichment scores for Nucleus and Endoplasmic Reticulum (ER; combination of ERM and ER lumen). However, it must be noted that when looking at different regions of tissue that are enriched for different cell types, the nuclear and ER enrichment scores change for each fluxmap (Fig. S4C).

- *One reason previous spatial transcriptomics analysis approaches don't look into the subcellular information is probably the result of the difficulty in handling the subcellular information of each molecule within each cell. Bento uses anndata to deposit the data but doesn't support subcellular information for each molecule within each cell. I would love to see how the author deals with this and, importantly, whether they can come up with a nice solution to encode both the subcellular information for each molecule and spatial information of each cell.*

Bento not only utilizes the AnnData object to store gene expression counts, it also stores molecular coordinates and segmentation boundaries e.g. nuclei and cell membrane and this is both described in our online documentation (<https://bento-tools.readthedocs.io/en/latest/howitworks.html#data-structure>) and is easily accessible via Python API functions including `bento.geo.get_points` (https://bento-tools.readthedocs.io/en/latest/api/bento.geo.get_points.html) and `bento.geo.get_shape` (https://bento-tools.readthedocs.io/en/latest/api/bento.geo.get_shape.html). More advanced data formatting is out of scope of this paper as we believe it is more useful to collaborate on a standard as a community than to write proprietary or highly customized formats that are difficult to work with outside of Bento. The recently published SpatialData (built on top of OME-TIFF, OME-NGFF, Zarr, and AnnData file formats) shows how difficult this is i.e. harmonizing molecules, images, shapes, multiple experiments and technologies, coordinate system registration, transformations etc. We are actively collaborating with SpatialData developers to use their data structure in a future release.

- *Another reason previous spatial transcriptomics analysis approaches don't look into the subcellular information is probably the fact that the measurement of subcellular information requires high spatial resolution and precision. At this moment, it is unclear what the spatial resolution is for each spatial transcriptomics technology. It will be important for the author to assess the level of spatial precision for each major spatial transcriptomics technology. Importantly, they should quantify the level of RNA diffusion during the imaging or sequencing library preparation step. Furthermore, it will be important to discuss the level of subcellular analysis that can be conducted for each technology, given their subcellular spatial resolution.*

Regarding the spatial resolution of technology platforms, we believe it is the responsibility of the authors/companies to build trust and provide quality metrics for the chemistry in their respective technologies. However, we have included a discussion on how molecule density, segmentation quality and target panel composition of each dataset affect application of RNAforest, RNAcoloc, and RNAflux analyses in our discussion section.

- *It is a nice idea to compare the data from APEX-seq with the MERFISH data. However, APEX-seq is based on bulk measurements. It will be very interesting to see whether the subcellular RNA distribution differs across different cell types, etc.*

We agree APEX-seq data is not quite single-cell resolution. In this paper we have applied RNAflux to evaluate subcellular RNA distribution of APEX-seq data across multiple cell types including bone osteosarcoma U2-OS cells (MERFISH) and iPSC-derived cardiomyocytes (Molecular Cartography) while the APEX-seq data itself is derived from human embryonic kidney HEK293 cells. We also hope to better understand how RNA localization is conserved across cell types, tissues, and organisms as Bento is adopted to analyze more spatial transcriptomics datasets in the future. To this end, we measured the enrichment of the organelle-specific APEX-seq genesets in the fluxmaps generated from breast cancer tissue (Xenium), and found that while RNAflux was capable of finding discrete subcellular regions, the ability of the APEX-seq genesets to corroborate compartment identity was variable. We posit that this is driven by both cell-type specific RNA enrichment in each organelle, as well as limited overlap between the Xenium targeted panel and the APEX-seq data generated by U2-OS. In the end, we utilized the APEX-seq data simply to corroborate the accuracy of RNAflux in the absence of

immunofluorescence co-stains. However, as platforms mature and increasingly add spatial proteomics capabilities in combination with transcriptomics, the need for using the APEX-seq genesets will be eliminated. We discuss the RNAflux and APEX-seq application to the breast cancer tissue dataset in the second to last paragraph in the section titled “RNAflux: Unsupervised semantic segmentation of subcellular domains in single cells”.

- *Tensor decomposition for compartment-specific colocalization: I don't follow the mathematical description in this section. Can you explain more about what the dataset tensor X is? And what is the rank-one 3-way tensors?*

We agree with the reviewers that tensor decomposition in the context of RNAcoloc needs better explanation. We have rewritten the methods section to be much more detailed including appropriate notation to describe tensor decomposition. See the updated method section titled “Tensor decomposition for compartment-specific colocalization” below:

We begin by calculating the CLQ for every pair of genes within each compartment of every cell. We structure our data as follows:

1. Cells set: Denote the set of cells as $C = \{c_1, c_2, \dots, c_n\}$, where c_i represents the i^{th} cell and n is the total number of cells.
2. Compartments set: Every cell has the same set of compartments, represented as $K = \{k_1, k_2, \dots, k_m\}$, where k_j is the j^{th} compartment within a cell.
3. Gene pairs set: The gene pairs are represented by $G = \{g_1, g_2, \dots, g_p\}$ where g_p is the p^{th} gene pair.

By computing the CLQ for every combination of cells in C , compartments in K , and gene pairs in G , we populate a 3-dimensional tensor X with dimensions corresponding to these sets.

We then apply non-negative parallel factor analysis (PARAFAC) as implemented in Tensorly[30] to reduce the dimensionality of our dataset and capture the underlying patterns. For tensor decomposition, we employed non-negative parallel factor analysis as implemented in Tensorly[30]. The tensor X is decomposed into the sum of R factors; each factor is a 3-dimensional tensor expressed as the outer product of three vectors: x_r (compartment loadings), y_r (cell loadings), and z_r (gene pair loadings). This is denoted as follows:

$$\hat{X}_R = \sum_{r=1}^R x_r \otimes y_r \otimes z_r$$

The optimal rank- R decomposition of X is determined by minimizing the error between X and the reconstructed tensor \hat{X} . We use the elbow function heuristic to select the best fit rank from a range of 2-12 factors. This approach seeks to balance the complexity of the model against the fidelity of reconstruction. Missing values in X are ignored when calculating the loss.

- *Make sure to use either capitalized or lowercase a/b/c in figures and figure captions consistently. Currently, all figures use lowercase, but captions use capital letters.*

Figure lettering has been capitalized.

- *Figure 1: What does the color in panel D mean? You may want to explicitly mention that the same set of colors from panel A is consistently used throughout the figure.*

Assuming the reviewer is referring to figure 2 instead of figure 1, we have added an indication in the figure 2 legend that the same colors are used consistently throughout the figure.

- *[Figure 1]: It is weird to see that although ANKRD52, CEP250, etc., are treated as "None" genes, they show enrichment in the cytoplasmic group. Also, please label the legend for the colormaps in panel H/I.*

All genes exhibit a distribution of localization patterns since RNA are not static. This means that the RNA localization of a given gene can be heterogeneous (have more than 1 dominant pattern) across a population of cells. Figure H reports the top 5 genes for each label sorted by label frequency. We have added legends to the colorbars in panel H/I as suggested.

- *Clustermap is known to have low performance for single-cell segmentation. Can you please try Baysor or Spateo for single-cell segmentation?*

While cell segmentation of the cardiomyocytes dataset can certainly be improved, we do not think it is necessary to evaluate the performance of different cell segmentation algorithms for illustrating the utility of Bento. Currently there is no consensus on a best performing method; performance just as much across datasets as between tools (Bering Fig 2, ComSeg Fig 2). Recent benchmarks (mutual information, IoU, cell type heterogeneity, jaccard index etc.) show no clear improvement of one method over another, including methods such as watershed, Cellpose, pciSeq, ComSeg, Bering, Clustermap, and Baysor. See the following benchmarks from recently published models Bering (<https://www.biorxiv.org/content/10.1101/2023.09.19.558548v1>) and CellSeg (<https://www.biorxiv.org/content/10.1101/2023.12.01.569528v1#:~:text=Yet%2C%20accurate%20cell%20segmentation%20can,explicit%20priors%20on%20cell%20shape>). In Bering Figure 2, ClusterMap is denoted as CM.

Bering Fig 2

ComSeg Fig 2

- The data shown in the last figure is interesting. However, the analysis for this dataset is rather shallow. For example, we can ask the following questions:

Can you find some genes that don't show expression changes before and after drug treatment but have a high degree of shifts in terms of subcellular distribution?

In panels F/G, you highlighted the depletion of RNA within the cytoplasm. Can we find genes that show the opposite pattern? What genes show high depletion in the nucleus after the Dox treatment?

We performed a spearman correlation to quantify the relationship between expression changes and subcellular distribution shifts and found no significant correlation. In updated Figure 5F, we would expect to see genes that are depleted from the nucleus (red vertex) after treatment to be closer to the yellow (ERM/OMM), blue (ER lumen) or green (cytosol) vertices. However no genes show shifts in localization that are also consistently expressed in the cardiomyocytes (circle size ~ fraction of cells, Fig. 5F). We have updated Figure 5 to include nucleus and ER enrichment score maps (Fig. 5E) and plots showing LAMP2 localization (Fig. 5G).

We added the following to the main text on cardiomyocyte RNAflux results:

There was no correlation when comparing the logFC in expression and the difference in nuclear composition of genes after treatment, indicating that localization towards the nucleus is not driven by transcript abundance (spearman $r=0.07$, $p=0.4944$).

Most genes only showed weak shifts in localization, similar to LAMP2 (Fig. 5G right).

Minor Comments

- Some language may need improvement. For example, I don't follow (in the section RNAflux: xxxx, last paragraph) why the computation of a gene composition vector will generate a spatial computation gradient across entire cells. In your figure 4a, the RNAflux embedding shows different colors, and the colors are generated from the top three principal components. Is it precise to call the color difference the computation gradient? I would suggest providing more details in that sentence.

We replace the term "spatial composition gradient" with "spatially coherent embedding". Additionally, we have revised the methods section to better describe the algorithm with appropriate mathematical notation.

- The font size for the CLQ score and spatial domains in Figure 4b is too small. A minimal font size of 6 is needed. Font sizes for Figure 5e are too small as well.

We thank the reviewer for the suggestion and we have corrected the font sizes.

Second round of review

Reviewer 1

I am satisfied with the author's revision and recommend publication of this work.

Reviewer 2

The authors addressed reviewers' comments well. The revised version is improved in quality. I have no further suggestions to make.

Reviewer 3

The authors largely addressed my comments and I don't have further comments